# Improving Thermal Stability of Polyurethane through the Addition of Hyperbranched Polysiloxane

**DOI:** 10.3390/polym11040697

**Published:** 2019-04-16

**Authors:** Shang-Hao Liu, Ming-Yuan Shen, Chen-Feng Kuan, Hsu-Chiang Kuan, Cing-Yu Ke, Chin-Lung Chiang

**Affiliations:** 1Department of Ammunition Engineering and Explosion Technology, Anhui University of Science and Technology, Huainan 232001, Anhui, China; u9414042@cmu.edu.tw; 2Department of Mechanical Engineering, National Chin-Yi University of Technology, Taichung 411, Taiwan; hbj678@gmail.com; 3Department of Food Beverage Management, Far East University, Tainan 744, Taiwan; cfkuan@mail.feu.edu.tw (C.-F.K.); hckuan@mail.feu.edu.tw (H.-C.K.); 4Green Flame Retardant Material Research Laboratory, Department of Safety, Health and Environmental Engineering, Hung-Kuang University, Taichung 433, Taiwan; n74731@gmail.com

**Keywords:** polyurethane (PU), siloxane, sol–gel technology, thermogravimetric analysis, thermal stability

## Abstract

Polydimethylsiloxane with hydroxy groups was functionalized to form functionalized polydimethylsiloxane, which subsequently underwent an addition reaction with isophorone diisocyanate to form the prepolymer. Next, 3-aminopropyltriethoxysilane (APTS) reacted with 3-glycidoxypropyltrimethoxysilane (GPTS) to produce bridged polysilsesquioxanes, and sol-gel technology was employed to form hyperbranched polysiloxane nanoparticles with hydroxy groups, APTS-GPTS, which was used as the additive. The hyperbranched polysiloxane and the prepolymer containing NCO functional groups then underwent an addition reaction to produce the hybrid materials. Fourier-transform infrared spectroscopy and ^29^Si nuclear magnetic resonance were used to characterize the structure of the polyurethane hybrid. Regarding thermal stability, after the hyperbranched polysiloxane nanoparticles was introduced, the integral procedural decomposition temperature increased from 348 °C for polyurethane matrix to 859 °C for the hybrid material. The results reveal that the thermal stability of the hybrid material substantially increased by approximately 247%.

## 1. Introduction

Polyurethane (PU) is a common high-performance polymer that features favorable mechanical properties, chemical resistance, and wear resistance. Polyurethane material is widely applied in automobile manufacturing, the textile industry, sporting equipment, shoe soles, and coating materials [1,2]. Despite its versatility, PU is not resistant to heat. For example, the mechanical properties of PU deteriorate rapidly when the temperature reaches 90 °C or higher. Furthermore, PU underdoes severe pyrolysis when the temperature exceeds 200 °C. Therefore, limiting applications of PU is critical to the development of polymer materials [3,4,5]. 

Numerous studies have investigated the siloxane structure, which is unique in terms of its thermal stability, hydrophobicity, weatherability, high gas penetrability, low toxicity, insulative properties, and excellent ultraviolet resistance. This structure can effectively enhance the heat resistance of polymer materials that have been mixed with siloxane, thereby expanding the application scope of polymer materials [6,7,8].

PU possesses excellent properties but has limited applications because it has poor thermal stability and easily combusts. By comparison, the main chain of siloxane is the Si-O bond, which has high bond energy and significant chain flexibility; thus, siloxane features superior thermal stability under high temperature [9,10]. In this study, the sol–gel technology was adopted to mix organic and inorganic materials to form a siloxane-PU hybrid material. Subsequently, Fourier-transform infrared spectroscopy (FTIR) was employed to identify the structure of the hybrid material, and solid-state ^29^Si nuclear magnetic resonance (NMR) spectroscopy was used to identify the condensation density of the Si-PU/APTS-GPTS hybrid material. The thermal stability and heat resistance of the developed hybrid materials were examined using thermogravimetric analysis (TGA) and the integral procedural decomposition temperature (IPDT).

## 2. Experiment

### 2.1. Materials

Diglycidyl ether of bisphenol A (DGEBA) with an epoxide equivalent weight (EEW) of 180 g eq^−1^ was obtained from Nan-Ya Plastics Corporation, Taipei, Taiwan. Diaminodiphenylmethane (DDM), which was used as curing agent for epoxy resin, was purchased from TCI Chemical Co, Tokyo, Japan. *N*,*N*-Dimethylbenzylamine (NDBA), 1,4-butanediol(1,4-BD), 3-aminopropyltriethoxysilane (APTS) and 3-glycidoxypropyltrimethoxysilane (GPTS) were purchased from Acros Chemical Co, Springfield Township, NJ, USA. Polydimethylsiloxane KF-6000 (PDMS) was purchased from Topco Technologies Corp, Taipei, Taiwan. Anhydrous stabilized tetrahydrofuran (THF) was obtained from Lancaster Co., Morecambe, Lancashire, UK. Isophorone diisocyanate (IPDI) and dibutyltin dilaurate (DBTDL) were purchased from Alfa Aesar CO., Shore Road, Heysham, LA3 2XY, UK. Hydrogen chloride (HCl) was purchased from ECHO Chemical Co., LTD, Toufen Chen, Miaoli, Taiwan.

### 2.2. Preparation of Si-PU

PDMS (18.24 g) with hydroxy groups (-OH) and epoxy (1.76 g) were placed in a 100-mL serum vial and underwent magnetic and mechanical agitation in a nitrogen environment at 80 °C. NDBA catalyst (0.2 g) was then added to the serum vial to react with the mixture for 6 h to form the functionalized polydimethylsiloxane (FPDMS). After the temperature was reduced to 40 °C, 8.88 g of IPDI and 20 g of FPDMS were placed in a four-neck flask in nitrogen environment and then underwent magnetic and mechanical agitation at 80 °C. Subsequently, DBTDL catalyst (0.3 g) was added to the four-neck flask to react, forming a prepolymer. After reacting for 2 h, the viscosity of the prepolymer was increased to a degree similar to that of maltose, and subsequently added 80 mL of THF solvent and 0.8 g of 1,4-butanediol chain extender to react for 1 h to form polyurethane containing silicon, Si-PU. After the viscosity increased, Si-PU was placed into to a Teflon mold. The mold was first placed in a vacuum oven for deaeration at 80 °C for 24 h and then placed in a circulating oven for 24 h at 80 °C. The finished product was then removed from the mold and placed at room temperature to cool, and completed the preparation of Si-PU. The reaction of Si-PU is presented in Scheme 1.

### 2.3. Preparation of APTS-GPTS

APTS (1.55 g) and GPTS (1.65 g) were added to a 100-mL serum vial with 80 mL of THF solvent for agitation; after reacting for 2 h, Solution A was obtained. Next, the THF solvent was added to 0.72 mL of deionized water and hydrogen chloride was instilled, and the pH value was adjusted to 4, thereby obtaining Solution B. Finally, Solution B was slowly instilled into Solution A and underwent a sol–gel reaction for 2 h at 60 °C to obtain APTS-GPTS, the reaction of which is shown in Scheme 2.

### 2.4. Preparation of Si-PU/APTS-GPTS

Prepolymer (28.88 g) was added to 80 mL of THF solvent and APTS-GPTS (19.8 g) was slowly instilled and left for 2 h to react to observe whether the viscosity had increased and the fluid level had decreased. After the viscosity had increased, the mixture was poured into a Teflon mold. The mold was first placed in the vacuum oven for deaeration for 24 h. Once the temperature was 80 °C, it was then placed in the cyclic oven for 24 h. Once the temperature had reached 80 °C, the finished product was then removed from the mold and cooled to room temperature, thereby completing the preparation of Si-PU/APTS-GPTS. The reaction of Si-PU/APTS-GPTS is shown in Scheme 3. 

### 2.5. Measurements

The FTIR spectra of the materials were recorded within 4000–400 cm^−1^ using a Nicolet Avatar 320 FT-IR spectrometer, from Analytical Instruments Brokers LLC, Golden Valley, MN, USA. Thin films were prepared by the solution-casting method. A minimum of 32 scans were signal-averaged with a resolution of 2 cm^−1^ in the 4000–400 cm^−1^ range. ^29^Si NMR was performed by a Bruker DSX-400WB, Germany. The samples were treated at 180 °C for 2 h and then ground into fine powder. The thermal degradation of composite was examined using a thermogravimetric analyzer (TGA) (Perkin Elmer TGA 7) from room temperature to 800 °C at a rate of 10 °C /min under an atmosphere of nitrogen. The measurements were made on 6–10 mg samples. Weight-loss/temperature curves were plotted. 

## 3. Results and Discussion

### 3.1. Characterization of Si-PU/APTS-GPTS Hybrid

The PDMS with –OH functional groups was used for a modification reaction with epoxy, and FTIR was used for structural characterization. Figure 1 shows the FTIR spectra of the PDMS with –OH, epoxy, and FPDMS, revealing that the PDMS had a large characteristic absorption peak of –OH at 3600–3200 cm^−1^ [11,12] and an Si–O–Si functional group at 1080 cm^−1^ [13]. The epoxy also exhibited a large oxirane ring at 910 cm^−1^ [14,15]. FPDMS was formed after a ring opening reaction. In the spectra, we observed the disappearance of the oxirane ring and the appearance of a C-O characteristic absorption peak at 1107 cm^−1^ [16]. These results reveal that the ring opening reaction was successful. 

Figure 2 shows the FTIR spectra of the prepolymer that was generated from the reaction between IPDI and FPDMS. In Figure 2, the –NCO functional group of IPDI was clearly observed at 2270 cm^−1^ [17], and the -OH functional group of FPDMS was observed at 3600–3200 cm^−1^ [11,12]; the two reacted to produce a prepolymer. The spectra revealed that the characteristic absorption peak of -NCO partially disappeared, and the remaining –NCO functional groups were beneficial to the next stage of the reaction. Additionally, the reaction between IPDI and FPDMS resulted in the appearance of characteristic absorption peaks of –NH at 3320 cm^−1^ [18], C–N at 1310 cm^−1^ [19], and C=O at 1700–1630 cm^−1^ [20]. These results confirm that the IPDI–FPDMS reaction had occurred. 

As shown in Figure 3, the characteristic absorption peak of CH_2_ stretching on the APTS spectrum appeared at 3050–2800 cm^−1^ [16,21], and those of N-H, Si–O–C, and Si–OEt were at 1640–1550 cm^−1^ [22,23], 1200–1000 cm^−1^ [24], and 1180 cm^−1^ [25], respectively. Moreover, the characteristic absorption peak of the oxirane ring group on the GPTS spectrum was at 910 cm^−1^ [14,15]. After the ring opening reaction between APTS and GPTS, the spectra clearly showed the disappearance of the oxirane ring and the appearance of a C–N characteristic absorption peak at 1310 cm^−1^ [19], revealing that APTS-GPTS underwent a successful ring opening reaction. Moreover, the APTS-GPTS spectrum retained the characteristic absorption peaks originally on the APTS and GPTS spectra, also confirming the success of the reaction. 

Figure 4 shows that the -NCO functional group of the prepolymer at 2270 cm^−1^ [17] and the -OH functional group of APTS-GPTS at 3600–3200 cm^−1^ [11,12] reacted. Because the -NCO functional group fully reacted with the -OH functional group, the -NCO functional group on the Si-PU/APTS-GPTS spectrum disappeared and the N-H characteristic absorption peak became more evident at 3320 cm^−1^ [18]; other functional groups originally on the prepolymer and APTS-GPTS spectra were also present. This demonstrated that the Si-PU/APTS-GPTS hybrid material had been successfully fabricated. 

### 3.2. Network Structure of Si-PU/APTS-GPTS Hybrid

Solid-state ^29^Si NMR spectroscopy can provide valuable and accurate information on the siloxane structure. Solid-state ^29^Si NMR spectroscopy was used to identify the structure of Si-PU/APTS-GPTS after it underwent the synthesis reaction. During the synthesis reaction, sol–gel technology was used to observe the hydrolysis- condensation level of the material. Because three trialkoxy groups (T) existed at one end of the APTS and GPTS, the structure of the APTS and GPTS was T-shaped. On the other end, the organic chain NH_2_ and epoxy underwent a ring opening reaction. The T group end of the APTS and GPTS further underwent hydrolysis-condensation to form a Si–O–Si network structure that displayed high stability. Based on the level of hydrolysis-condensation, the absorption peak of the mono-substituted T was at −45~−48 ppm and was defined as the T^1^ structure [26,27], the absorption peak of di-substituted T was at -56~−62 ppm and was defined as the T^2^ structure [26,28], and the absorption peak of tri-substituted T was at −66 to −69 ppm and was defined as the T^3^ structure [26,27]. 

Figure 5 shows the solid-state ^29^Si NMR spectrum of Si-PU/APTS-GPTS hybrid material. As shown, the structure of Si-PU/APTS-GPTS consisted of T-shaped structures, with the T^3^ structure being the primary structure. Through peak separation, the integral areas of the T^1^, T^2^, and T^3^ structures were obtained. Accordingly, the structure of the siloxane composite comprised 20.1% of the T^1^ structure, 34.1% of the T^2^ structure, and 46.0% of the T^3^ structure. Subsequently, the condensation density (Dc%) of Si-PU/APTS-GPTS was calculated as 75.4% using Equation (1) [29]. The high Dc% indicated a compact network structure in the material, that the Si-O-Si bonds had formed a favorable network structure, and that the Si-O bond possessed high bond energy. This could effectively enhance the thermal stability of the hybrid material. The results are shown in Table 1.
(1)$$\mathrm{Dc}\left(\%\right)=\frac{1\times \left(\%areaT^{1}\right)+2\times \left(\%areaT^{2}\right)+3\times \left(\%areaT^{3}\right)}{3}$$

### 3.3. Thermogravimetric Analysis

Based on the microbalance principle, a thermogravimetric analyzer records the weight loss of samples according to changes in the physical and chemical properties of substances as the temperature and time increase, thereby determining the thermal stability of materials. In the TGA, the heating rate was set to 20 °C/min and was measured in a nitrogen atmosphere. When APTS-GPTS was added to Si-PU at different contents, the weight loss changed as the temperature increased. The results are shown in Figure 6 and Figure 7 and Table 2. 

Figure 6 and Table 2 show the changes in the decomposition curves. The temperature (T_d5_) of pristine PU was 273 °C when the weight loss was 5%. After APTS-GPTS was added to Si-PU, and as the concentration of APTS-GPTS increased, T_d5_ increased considerably to 330 °C (Si-PU/APTS-GPTS 40% composite) when the weight loss of the hybrid material was 5%. As the concentration of APTS-GPTS increased, the char yield increased, particularly at 800 °C. The char yield increased from 0.7 wt% for pristine PU to 24.7 wt% for the Si-PU/APTS-GPTS 40% composite. This can be attributed to the surface migration of Si in the APTS-GPTS structure when APTS-GPTS underwent thermal decomposition [30,31]. The surface migration generated a compact SiO_2_ structure to protect the internal part of the material. The benzene ring contributed to the char yield; therefore, the char layer featured oxidation resistance to prevent combustion. Figure 6 shows the large differences in weight retention between different samples, revealing that APTS-GPTS increased the thermal stability of Si-PU.

The derivative thermogravimetric curves are displayed in Figure 7. As shown, as the temperature changed, the temperature of maximum degradation rate of pristine PU was 343 °C. As the concentration of APTS-GPTS increased, the temperature of maximum degradation rate increased to 508 °C (Si-PU/APTS-GPTS 40%). Moreover, maximum degradation rate for pristine PU was −36 wt %/min. After adding APTS-GPTS, it effectively slowed down to −9.6 wt%/min and greatly slowed down the rate of thermal degradation. The above proved that the addition of APTS-GPTS nanoparticles could effectively improve the thermal stability of hybrid materials. 

### 3.4. Integral Procedural Decomposition Temperature (IPDT)

The TGA data were used to plot a diagram as Figure 8. The area under the decomposition curve was obtained through integration in following picture, and was then substituted into Equation (2) to calculate the IPDT, which was used to evaluate the thermal stability of the composite materials [32]. The two major factors that influenced the calculation results were the initial decomposition temperature and char yield. High values for these two factors indicated favorable thermal stability, heat resistance, and IPDT, whereas low IPDT indicated poor overall thermal stability of the materials.

The calculation was performed using the following equation:
A* = (S_1_ + S_2_)/(S_1_ + S_2_ + S_3_)
K* = (S_1_ + S_2_)/S_1_
where T_i_ is the initial experimental temperature, and T_f_ is the final experimental temperature. The IPDT can be obtained by incorporating the calculated values of T_i_, T_f_, S_1_, S_2_, and S_3_ into the Equation (2).
IPDT (°C) = A* × K* × (T_f_ − T_i_) + T_i_(2)

Figure 9 and Table 2 show that the IPDT of pristine PU was 348 °C, which increased to 859 °C after APTS-GPTS was added to 40%; that is, the IPDT of the hybrid material was greater than that of the pristine PU by 511 °C, and the thermal stability substantially increased by approximately 247%. These results demonstrate that adding APTS-GPTS enhanced the thermal stability of PU hybrid materials, and such high thermal stability could be attributed to the superior degradation resistance of APTS-GPTS at high temperature. 

### 3.5. Experimental and Calculated Data

Based on the substitution of the TGA curves, calculated data were obtained using Equation (3). The calculated data were then compared with the experimental data to understand whether the data exhibited a positive deviation when APTS-GPTS was added to polymer materials. A positive deviation denoted a favorable interaction between the organic and inorganic phase. The equations for the calculated data are as follows [33]:
Calculated data 10% = Si-PU*90% + APTS-GPTS*10%
Calculated data 20% = Si-PU*80% + APTS-GPTS*20%
Calculated data 30% = Si-PU*70% + APTS-GPTS*30%
Calculated data 40% = Si-PU*60% + APTS-GPTS*40%

The difference between the experimental and calculated TGA curves revealed a strong interaction between the inorganic phase of APTS-GPTS and the organic phase of Si-PU (Figure 10). The TGA curve of the Si-PU/APTS-GPTS 40% composite showed a char yield higher than that of the calculated TGA curve by 8.3%, demonstrating that the experiment data in this study produced satisfactory thermal stability [34]. 

Changes in the initial decomposition temperature and char yield were examined, as shown in Table 3. The experimental decomposition temperature (T_d5_) of the Si-PU/APTS-GPTS 40% composite was higher than the calculated temperature by 88 °C [34]. The initial decomposition temperatures for the four samples were considerably higher than the calculated temperatures. Furthermore, the char yields of experimental data all exhibited a positive deviation. These results reveal that the combination of organic and inorganic phases effectively increased the initial decomposition temperature and the char yield. Therefore, the satisfactory results were not the effect of a single material, but represented a synergistic effect of all the materials in the composite.

## 4. Conclusions

This study conducted the structural identification of hybrid materials synthesized with APTS-GPTS by using FTIR and ^29^Si NMR. The identification verified that introducing APTS-GPTS to the polymer materials successfully prepared the Si-PU/APTS-GPTS hybrid materials. Regarding the thermal properties, this study revealed that adding siloxane to siloxane-PU effectively enhanced thermal stability and heat resistance. The thermal stability was then measured using IPDT, and the results show that the thermal stability of the materials increased considerably. The experimental and calculated data exhibit a positive deviation, demonstrating that APTS-GPTS interacted with the polymer materials. The results of this study reveal that APTS-GPTS effectively improved the thermal stability of PU.

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
