# Peer review of "Improving Thermal Stability of Polyurethane through the Addition of Hyperbranched Polysiloxane"

_polymers, 2019, doi:10.3390/polym11040697_

Round 1

Reviewer 1 Report

The polyurethane was modified with polydimethylsiloxane with the aim to enhanced thermal stability of the prepared hybrid materials. The degradation temperature changed from 348 ºC for polyurethane matrix to 859 ºC for polyurethane modified with polydimethylsiloxane. Before the published this paper mayor revision are requested.

Mayor revision:

1.      Please specify the method for determination of degradation temperature. I do not understand from your TGA curves how you find 859 ºC for degradation temperature of hybrid materials. The onset of first degradation step should be taken as the degradation temperature of each materials if not the maximum temperature of first derivative TGA curve peak for each curve: In any case this temperature is equal 859 ºC.

Minor revision:

1. The molecules in each scheme should not be deform. Please correct this along the manuscript.

Author Response

Dear Reviewers:

Firstly, we would like to thank you for your comments and we have corrected the necessary text. The changes in the text have been highlighted in red.

Reviewer #1:

Q1: Please specify the method for determination of degradation temperature. I do not understand from your TGA curves how you find 859 ºC for degradation temperature of hybrid materials. The onset of first degradation step should be taken as the degradation temperature of each materials if not the maximum temperature of first derivative TGA curve peak for each curve: In any case this temperature is equal 859 ºC.

A1: Thanks for your kind suggestions. In this study, we used integral procedural decomposition temperature (IPDT) to evaluate the thermal stability of the composite materials. Figure 8 and Table 2 show that the IPDT of pristine PU was 348 ºC, which increased to 859ºC after APTS-GPTS was added to 40%; that is, the IPDT of the hybrid material was greater than that of the pristine PU by 511 ºC, and the thermal stability substantially increased by approximately 247%. In table 2, we also used Td5 to describe the thermal stability of the materials, which was initial decomposition temperature.

Q2: The molecules in each scheme should not be deform. Please correct this along the manuscript.

A2: Thanks for your kind suggestions. We have redrawn the reaction schemes to avoid the deformation.

Reviewer 2 Report

The corresponding author of this article  published in the past a work which should mentioned in this paper and included in the references, due to partly similar approach:

SYNTHESIS, CHARACTERIZATION, AND THERMAL PROPERTIES OF BRIDGED POLYSILSESQUIOXANES–MOLECULAR NANOCOMPOSITES Chin-Lung Chiang and Ri-Cheng Chang Department of Industrial Safety and Health Hung-Kuang University, 433, Sha-Lu, Taiwan

The integral procedural decomposition temperature (IPDT) proposed by Doyle and used in this work (Line 211) in my opinion is not a reliable parameter for real thermal stability, because the impact of char content on its value results in obtaining values taken by authors for thermal stability above 600 â„ƒ when in fact organic polymers are fully decomposed. 

Reliable thermal stability is shown in this work by direct TGA results, like Figs 6 and 7.

The reaction schema 1  contains errors, some hydrogen atoms are missing 

Why do you assume (scheme 1) that the secondary hydroxyl groups in PPDMS do not react with the diisocyanate and the prepolymer has a linear structure with two reactive NCO groups?

Bifunctional and linear prepolymer in reaction with 1,4 BD could not produce branched/crosslinked SI-PU.

The free NCO content in the prepolymer should be experimentally estimated in order to add a proper amount of the chain extender. 

LIne 43

 "The PU used in this study possesses excellent properties but has limited applications because it  has poor thermal stability and combustible easily" - How do you know that?  Physical and tensile properties were not estimated. The greatest problem at fire retardancy of polyurethanes is to not spoil the good properties of PU by excessive amount of  the fire retardant.

Without controlling of the tensile properties of the composite you cannot optimize the fire retardant content taking into account only the thermal stability which might be the best for a very weak composition.

Author Response

Dear Reviewers:

Firstly, we would like to thank you for your comments and we have corrected the necessary text. The changes in the text have been highlighted in red.

Reviewer #2:

Q1: The corresponding author of this article published in the past a work which should mentioned in this paper and included in the references, due to partly similar approach:

   SYNTHESIS, CHARACTERIZATION, AND THERMAL PROPERTIES OF BRIDGED POLYSILSESQUIOXANES–MOLECULAR NANOCOMPOSITES Chin-Lung Chiang and Ri-Cheng Chang Department of Industrial Safety and Health Hung-Kuang University, 433, Sha-Lu, Taiwan

A1: Thanks for your kind suggestions. We have added the article into the references.

Q2: The integral procedural decomposition temperature (IPDT) proposed by Doyle and used in this work (Line 211) in my opinion is not a reliable parameter for real thermal stability, because the impact of char content on its value results in obtaining values taken by authors for thermal stability above 600 â„ƒ when in fact organic polymers are fully decomposed. 

   Reliable thermal stability is shown in this work by direct TGA results, like Figs 6 and 7.

A2: Thanks for your professional suggestions. IPDT is only auxiliary tool to evaluate thermal stability of the materials. We have detailed discussion about thermal stability in Figs 6 and 7.

Q3: The reaction scheme 1 contains errors, some hydrogen atoms are missing 

A3: Thanks for your kind suggestions. We have corrected the mistake about scheme 1.

Q4: Why do you assume (scheme 1) that the secondary hydroxyl groups in PPDMS do not react with the diisocyanate and the prepolymer has a linear structure with two reactive NCO groups?

A4: Thanks for your kind suggestions. Secondary hydroxyl groups of PPDMS have steric hindrance, and they are not so reactive compared with primary hydroxyl groups of PPDMS. It is the reason that we assume that that the secondary hydroxyl groups in PPDMS do not react with the diisocyanate. We use stoichiometric ratio 2:1 for the diisocyanate to PPDMS and suggested the prepolymer has a linear with two reactive NCO groups.

Q5: Bifunctional and linear prepolymer in reaction with 1, 4 BD could not produce branched/crosslinked SI-PU.

A5: Thanks for your kind suggestions. We wholly agreed with your ideas about the structure of the prepolymer and corrected the scheme 1. We have redrawn the structure of Si-PU which is linear.

Q6: The free NCO content in the prepolymer should be experimentally estimated in order to add a proper amount of the chain extender. 

A6: Thanks for your kind suggestions. We wholly agreed with your professional advice about free NCO content measurement to determine a proper amount of chain extender. We use stoichiometric ratio for the reactants to calculate the amount of chain extender in this study. We will set up the experimental equipment to estimate free NCO content for next paper.

Q7: Line 43 "The PU used in this study possesses excellent properties but has limited applications because it has poor thermal stability and combustible easily" - How do you know that?  Physical and tensile properties were not estimated. The greatest problem at fire retardancy of polyurethanes is to not spoil the good properties of PU by excessive amount of the fire retardant.

A7: Thanks for your kind suggestions. We just describe common characteristics of PU in the introduction section. It is the motivation for the work. Maybe we use the words `` in this study``, they will let the readers confuse. We will delete the words.

Q8: Without controlling of the tensile properties of the composite you cannot optimize the fire retardant content taking into account only the thermal stability which might be the best for a very weak composition.

A8: Thanks for your kind suggestions. We have measured tensile properties to determine the contents of flame retardants in next paper. We are preparing the manuscript.

Round 2

Reviewer 1 Report

Accepted as is.